# The Role of Urine F2-isoprostane Concentration in Delayed Cerebral Ischemia after Aneurysmal Subarachnoid Haemorrhage—A Poor Prognostic Factor

**DOI:** 10.3390/diagnostics11010005

**Published:** 2020-12-22

**Authors:** Karol Wiśniewski, Marta Popęda, Bartłomiej Tomasik, Michał Bieńkowski, Ernest J. Bobeff, Ludomir Stefańczyk, Krzysztof Tybor, Marlena Hupało, Dariusz J. Jaskólski

**Affiliations:** 1Department of Neurosurgery and Neurooncology, Barlicki University Hospital, Medical University of Lodz, Kopcińskiego 22, 90-153 Lodz, Poland; ernestbobeff@gmail.com (E.J.B.); krzysztof.tybor@umed.lodz.pl (K.T.); marhup2@gazeta.pl (M.H.); dariusz.jaskolski@umed.lodz.pl (D.J.J.); 2Laboratory of Translational Oncology, Intercollegiate Faculty of Biotechnology, Medical University of Gdańsk, Dębinki 1, 80-211 Gdańsk, Poland; marta.popeda@gmail.com; 3Department of Biostatistics and Translational Medicine, Medical University of Lodz, 15 Mazowiecka St., 92-215 Lodz, Poland; bartektomasik@gmail.com; 4Department of Pathomorphology, Medical University of Gdańsk, Smoluchowskiego 17, 80-214 Gdańsk, Poland; michal.bienkowski@gmail.com; 5Department of Radiology—Diagnostic Imaging, Medical University of Lodz, Kopcińskiego 22, 90-153 Lodz, Poland; ludomir.stefanczyk@umed.lodz.pl

**Keywords:** F2-isoprostane, delayed cerebral ischemia, risk factors, aneurysmal subarachnoid hemorrhage

## Abstract

**Background:** The pathophysiology of delayed cerebral ischemia (DCI) remains unclear. One of the hypotheses suggests that reactive oxygen species play a role in its onset. Thus, we studied F2-isoprostanes (F2-IsoPs)—oxidative stress biomarkers. Our goal was to improve the early diagnosis of DCI in a non-invasive way. **Methods:** We conducted a prospective single center analysis of 38 aneurysmal subarachnoid hemorrhage patients. We assessed urine F2-IsoP concentration using immunoenzymatic arrays between the first and fifth day after bleeding. A correlation between urine F2-IsoP concentration and DCI occurrence was examined regarding clinical conditions and outcomes. **Results:** The urine F2-IsoP concentrations were greater than those in the control groups (*p* < 0.001). The 3rd day urine F2-IsoPs concentrations were correlated with DCI occurrence (*p* < 0.001) and long term outcomes after 12 months (*p* < 0.001). **Conclusions:** High levels of urine F2-IsoPs on day 3 can herald DCI.

## 1. Introduction

Worldwide, aneurysmal subarachnoid hemorrhage (aSAH) and its complications kill or seriously debilitate about 1.2 million people annually [1]. The most dangerous complication of aSAH is cerebral vasospasm (CVS), leading to delayed cerebral ischemia (DCI). The concept of CVS is complex; it can be diagnosed both angiographically and clinically. The narrowing of the contrast column in major cerebral arteries (i.e., radiologic or angiographic vasospasm) is detected on angiograms in about 70% of patients. In theory, cerebral vasoconstriction should lead to generalized ischemia. Nonetheless, the resultant deterioration of the clinical status and neurological deficits (i.e., clinical vasospasm) can be observed in up to 25% of patients. Thus, the causal link between angiographic and clinical CVS may be often not observed in clinical practice. DCI is a broad term encompassing both symptomatic vasospasm and neurological deficits. Its pathophysiology is multifactorial and not entirely clear [2,3], exceeding the simple causal link between vasospasm and infarction. In each case of DCI, a number of aspects should be considered, including early brain injury, microcirculatory constriction, microthrombosis, cortical spreading depolarization, blood–brain barrier disruption, dysregulated autoregulation, and delayed cell apoptosis.

DCI most frequently occurs between five and seven days post-aSAH, while never before the third day [1]. Experimental studies have shown that the risk of DCI correlates with clinical SAH severity and radiographic blood load [4]. 

A suspicion of DCI is raised in patients presenting with symptoms of confusion or decreased level of consciousness with or without focal neurologic deficits. Such cases are verified with angiography or transcranial Doppler [5,6]. In contrast, no reliable diagnostic tool exists for evaluating patients who are not clinically assessable (due to unconsciousness)**.** The most plausible theory on the pathogenesis suggests that the subarachnoid blood acts as a nidus of reactive oxygen species (ROS) and inflammatory factors, which promote the peroxidation of membrane lipids of endothelial cells and proliferation of smooth muscles cells leading to DCI [7]. Current guidelines recommend only oral nimodipine, which improve the neurologic outcomes, albeit not counteracting the DCI itself [8]; the research on a new drug is ongoing [9].

We propose that ROS production plays a crucial role in the onset of DCI. F2-isoprostanes (F2-IsoPs) are postulated as oxidative stress markers, since their concentration is correlated with ROS levels [10]. These compounds are formed as a result of free-radical peroxidation, a continuous process, occurring both in physiological and pathological conditions, involving the oxidation of polyunsaturated fatty acids present in cellular membranes [11]. In addition, F2-IsoPs mediate oxidative damage through their vascular, inflammatory, and mitogenic activity [12]. Among all identified isoprostanes, the strongest vasoconstricitive activity was observed for 8-iso PGF2a [13].

Here, we present a validation analysis of our previous report in an attempt to improve on the non-invasive diagnosis of DCI [14].

## 2. Materials and Methods

### 2.1. Patients

We performed a prospective analysis of consecutive aSAH patients treated at the Department of Neurosurgery, Medical University of Łódź, between June 2017 and January 2019. The control group consisted of 13 healthy volunteers (7 women and 6 men). The inclusion and exclusion criteria for both groups are listed in Table 1.

Chronic organ failure was listed among the exclusion criteria as it may affect the metabolism of isoprostanes (free F2-IsoPs are primarily eliminated by kidneys and to a lesser extent by liver; in addition, an association between F2-IsoPs and chronic heart failure was reported) [15,16].

### 2.2. Clinical Assessment

Every time subarachnoid hemorrhage (SAH) was suspected, we used computed tomography (CT) to confirm the diagnosis. In all cases with SAH, the presence of an aneurysm was verified on CT angiography (CTA) or digital subtraction angiography (DSA). The severity of patients’ condition was assessed according to the Hunt and Hess (H/H) scale, while the extent of hemorrhage was assessed according to the Fisher scale [17,18]. 

DCI was surmised in patients with neurological deterioration (confusion or decreased level of consciousness by at least 1 point on the Glasgow Coma Scale, with or without focal neurologic deficits, lasting for at least 1 hour) after excluding other causes of neurologic deficits. Each time we suspected DCI, we performed transcranial Doppler (TCD) in search of radiological signs of vasospasm. We examined the flow velocity in the middle cerebral artery (MCA) and internal cerebral artery (ICA). If the flow velocity in MCA exceeded 120 cm/s and the Lindegaard ratio (MCA low velocity/ICA flow velocity) was greater than 3, we performed a digital subtraction angiography (DSA). In our opinion TCD is not sufficient as a single screening measure. DSA is the gold standard and offers the possibility of endovascular treatment. 

Subsequently, the patients were examined by neurosurgeons blinded to experimental laboratory results at discharge and after 1 and 12 months (in parallel to routine out-patient clinic follow-up). The outcomes were measured according to Glasgow Outcome Scale (GOS) and the modified Rankin Scale (mRS) [19,20]. Routine checkups included the full examination (except for the cases of patient’s death, when we spoke with the family on the phone).

### 2.3. Specimen Collection

From the study group, 5–10 mL of morning urine samples (at least 8 h of fasting) were collected between the 1st and 5th day after aSAH (date of bleeding and surgery is regarded as day 0). The samples were centrifuged for 3 min at 3500× *g*, snap frozen in liquid nitrogen, and stored at −80 °C until analysis. In the control group (volunteers), urine samples were collected once, and processed in the same way.

### 2.4. Detection of Free form of F2-IsoPs in Urine

There were 181 samples in the study group (out of the intended 190, 9 samples were lost due to collection or storage error) and 13 samples in the control group available for measurements. We used a STAT-8-Isoprostane ELISA Kit (Cayman Chemical, Ann Arbor, MI, USA) to detect the free form of F2-IsoPs in urine. We conducted the analysis according to the manufacturer’s protocol. Before analysis, the samples were thawed and diluted 10-fold. In parallel, we conducted creatinine analysis in the same samples. To quantified creatinine, we used a Creatinine (urinary) Colorimetric Assay Kit (Cayman Chemical) and performed the analysis according to the manufacturer’s protocol. We used a 10-fold dilution of samples. Synergy 2 Multi-Mode Reader (BioTek Instruments, Inc., Winooski, VT, USA) and the dedicated software were used for plate readings. We normalized the concentration of F2-IsoPs per 1 mg of creatinine. 

### 2.5. Statistical Analysis

#### Power and Sample Size Analysis

The power and sample size analysis was performed for our primary aim, which was the assessment of urinary F2-isoprostane concentration as a prognostic factor for DCI development. On the basis of our previously published results [14], we sought a validation set showing a superiority of at least 0.25 in the area under the receiver operating characteristic curve (AUC ROC) against a value of 0.50 (assumed as a null hypothesis) with a statistical power of 80% and a type 1 error probability < 0.05. The calculation yielded a required set of 38 patients with balanced group counts.

Nominal variables are given as numbers with corresponding percentage. To test associations between categorical variables, we used chi-square tests with appropriate corrections. Continuous variables are presented as means with standard deviations or medians with interquartile range (25–75%), depending on the data distribution. For pairwise comparisons of continuous variables, *t*-test, or Mann–Whitney *U* test were used. Cohen’s d was used to assessed effect size. For multi-group comparisons, one-way analysis of variance (ANOVA) was used with post hoc comparisons performed with Tukey’s test.

Correlation between variables was analyzed using Kendall rank correlation.

The areas under the receiver operating characteristic curves (AUC ROC) were employed to illustrate the diagnostic ability and the discriminative performance of the analyzed markers as a predictor of DCI onset or poor clinical outcome (GOS score 1–3 after 12 months from surgery, mRS score 3–5 after 12 months from surgery). Youden’s method was employed to calculate optimal cutoff points.

In the multivariate analysis, binomial logistic regression was performed. Backward stepwise elimination of variables was implemented for feature elimination.

For all analyses, two-sided tests were used and *p*-values <0.05 were considered statistically significant. Statistical analysis was conducted using STATISTICA 13.1 (TIBCO Software, Palo Alto, CA, USA) and R version 3.6.2 (R Foundation for Statistical Computing, Vienna, Austria) with packages stats, car, pROC, glm2, and ggplot2 [21,22,23,24,25].

## 3. Results

### 3.1. The Patients’ Clinical Condition

A total of 60 patients were screened for eligibility and 38 were included (study flow-chart is presented in Figure 1). 

The clinical and laboratory data for all included patients are summarized in Table 2.

Among the 60 patients, there were 5 with H/H grade 4 hemorrhage who could not be included due to accompanying conditions (multiple aneurysms in 2 cases, coagulation disorder in 2 cases, and liver failure in 1 case).

DCI was diagnosed in 19 patients between 4 and 8 days after SAH. The diagnosis was made by two neurosurgeons and a radiologist.

### 3.2. Urine F2-IsoP Levels 

The levels of urine F2-IsoPs in the control group (*n* = 13) ranged between 5.8 and 8.8 pg/mg creatinine (mean 7.1 ± 0.9 pg/mg creatinine). The values in all aSAH patients were significantly greater than in controls (Table 3), both when analyzed cumulatively and for each day separately (*p* < 0.05 for all the comparisons).

Subsequently, we investigated the relation between F2-IsoP levels and clinical markers (including age, sex, aneurysm location, Hunt and Hess, Fisher, GOS, mRS). To this end, we dichotomized the quality of life scales according to the need for assistance in everyday life (i.e., GOS scores 4–5 vs. 1–3 and mRS scores 0–2 vs. 3–6). We noted that F2-IsoP levels on day 3 were associated with DCI occurrence (*p* = 0.007) and poorer clinical condition after 1 month (*p* = 0.024) and 12 months (*p* = 0.008) (Figure 2A–C). Next, we verified the predictive value of this marker with receiver operating characteristic (ROC) curves (Figure 2D–F).

In the backward stepwise logistic regression models, we investigated which combination of the clinical features provided the highest prognostic value regarding the occurrence of DCI. 

Receiver operating characteristic curves based on backward stepwise logistic regression showed that the best model for DCI prediction on the basis of the clinical markers incorporated only Hunt–Hess grade (AUC 0.733, 95% CI: 0.572–0.894) (Figure 3A). When we built a model with Hunt–Hass grade and F2-IsoP levels on day 3, we received even better results. Thus, the combination of F2-IsoPs on day 3 with this clinical scale turned out to be a very good outcome predictor (AUC 0.833, 95% CI: 0.703–0.964, *p* = 0.043) (Figure 3B). Notably, the differences remained significant after fivefold cross validation. 

Finally, we investigated the association of F2-IsoP levels with clinical conditions such as acute hydrocephalus and intracerebral hemorrhage (ICH). 

We noted eight cases with acute hydrocephalus (mean F2-IsoP urine level of all post-hemorrhage days was 11.7 ± 0.5 pg/mg creatinine). The mean level of urine F2-IsoPs in the acute hydrocephalus group was elevated compared with controls, but lower than in the DCI group. 

In five cases with intracerebral hemorrhage (mean F2-IsoP urine level of all post-hemorrhage days was 11.8 ± 0.3 pg/mg creatinine), the level was higher than in controls but lower than in the DCI group.

## 4. Discussion

In this study, we aimed to validate our previous report on urine F2-IsoP concentration in patients after aSAH [14]. To this end, we performed a single-center prospective analysis in a new group of patients.

DCI is a severe complication of SAH, which increases the risk of mortality by 1.5–3 times [26,27]. The potential consequences of DCI include cerebral infarction, poor functional outcome, and death [28,29,30]. Moreover, it has recently been identified as the most important predictor of neuropsychological deficits [31], while (in line with the improving aSAH survival rates) the functional outcomes, including the neuropsychological condition, have become the chief concern. Crucially, the detection of DCI in aSAH patients in poor clinical condition is challenging. In unconscious patients, up to 30% of DCI lesions are clinically silent [32,33]. For such cases, brain tissue oxygen tension monitoring (PbtO_2_) and cerebral microdialysis (CMD) have been proposed to detect ischemia and metabolic derangements, but these methods are invasive, expensive, and not readily available [34]. Undeniably, a reliable diagnostic tool would be beneficial since early identification of DCI may give the chance for effective treatment. 

The pathophysiology of DCI is complex and vastly unexplored [3]. Its development primarily depends on the amount of extravasated blood. It is also associated with older age, impaired consciousness (≥III Hunt–Hess grade), hyperglycemia, hypertension, hypovolemia, and tobacco use [35]. Among the several theories on DCI pathophysiology, it was proposed that the presence of subarachnoid hemoglobin leads to the release of reactive oxygen species (ROS). This results in the peroxidation of membrane lipids of endothelial cells and proliferation of smooth muscles cells [8,36]. Cell redox signaling, modified by ROS, regulates the vascular tone through various mechanisms, known as *redox switches*, leading to endothelial dysfunction and vasoconstriction. One of the *redox switches* relates to prostanoid synthesis (PGIS). As a key enzyme of PGIS, cyclooxygenase is activated at low peroxide levels. Oxidative stress (a condition with high peroxide levels) promotes isoprostane formation and inhibits PGIS synthesis [28,31]. Isoprostanes are considered to be oxidative stress markers, since their concentration directly correlates with that of ROS. The types of formed isoprostanes and their proportions depend on oxygen tension and glutathione concentration [37].

In the literature, elevated F2-IsoPs levels were reported in cerebral pathologies including chronic neurodegenerative diseases, including Alzheimer’s disease, Huntington’s disease, Parkinson’s disease, and amyotrophic lateral sclerosis [38,39,40,41,42]. In addition, Millard et al. described an association between higher DCI levels of F2-IsoPs and poorer executive function during aging [43]. Their role was also studied in a small group of aSAH patients (*n* = 15) with no observed relation to GOS, Fisher Scale, or Hunt and Hess grade scores [44]. Nonetheless, the sample size precludes any specific conclusions. 

In line with the multifactorial DCI mechanisms, not only do isoprostanes promote vasoconstriction and microvasospasm, but they also cause oxidative damage to the cell membrane or organelles. These compounds affect the fluidity and integrity of phospholipid membranes by changing the intermolecular interactions, leading to cell apoptosis and early brain injury. In addition, isoprostanes may stimulate platelet aggregation, promoting microthrombi formation, as well as induce cell cycle progression. Despite their clinical significance, no specific isoprostane receptor or the related intracellular pathway has been documented to date.

The choice of urine as the biomarker source was based on the properties of isoprostanes. They have a short blood half-life of about 16 minutes and are cleared primarily by the kidneys. In urine, the concentration 8-iso-PGF2α was shown to remain stable even after 5-day incubation of the sample at 37 °C [45]. This makes urine the preferable non-invasive source for biomarker analysis.

The levels of urine F2-IsoPs in our control group (*n* = 13) were relatively stable, ranging from 5.8 to 8.8 pg/mg creatinine (mean 7.1 pg/mg creatinine ± 0.9). Albeit low compared with the literature data [46,47], there results were similar to those presented in our pilot study [14]. As previously observed, F2-IsoP levels were significantly higher in aSAH patients. It is associated with high oxidative stress that arises after aSAH, created by extravasated blood. The peak levels of urine F2-IsoPs in aSAH patients, especially those in a very poor Hunt and Hess grade (III), were high, which could be responsible for severe local damage to the brain tissue, since F2-IsoPs caused vasoconstriction and platelet aggregation. We also observed that urine F2-IsoP levels were higher after posterior circulation aneurysms rupture, probably because those aSAHs reached higher in the Fisher scale, mostly grades 3 and 4 (Table 3). 

It should be also mentioned that urine isoprostane concentration in people over 60 years old was higher than those in younger patients (Table 3). We believe that in older patients the antioxidant capabilities of endogenous intracellular enzymatic antioxidants such as superoxide dismutase (SOD) and glutathione peroxidase (GSH-Px) is reduced and thus the balance shifts in favor of oxidative stress and isoprostane formation [48]. In the brain compartment, ROS production is predominantly countered by SOD (which protects against superoxide radicals), GSH-Px, and catalase (which protects against hydrogen peroxide); however, catalase appears to play a lesser role in the brain as compared to other organs [49].

We noted that mean urine F2-IsoP level in the DCI group was higher than in other conditions with severe acute brain injury such as the acute hydrocephalus group and the intracerebral hemorrhage group (17.5 ± 1.9 pg/mg creatinine vs. 11.7 ± 0.5 pg/mg creatinine vs. 11.8 ± 0.3 pg/mg creatinine, respectively).

Our results show that F2-isoP levels on day 3 after aSAH were significantly associated with the risk of DCI and worse long-term outcomes. These results are concordant with the findings reported in our pilot study [14]. All the ROC curves had area under the curve (AUC) >0.8, indicating the potential clinical value of these markers.

The limitation of our work is the single center setting; thus, the possibility of selection bias and measurement of random bias cannot be excluded. Secondly, several samples could not be measured due to improper material collection or storage. Finally, immunoassay kits to quantify urine 8-iso-prostaglandin F2a, despite its simplicity and cost-effectiveness, may be associated with cross-reactivity with other isoprostane isomers [50,51,52]. The extent of the biological activity of the isoprostanes remains to be determined because of limitations in the availability of individual isomers. However, the 8-iso-PGF2α isomer is a potent constrictor of smooth muscle in vitro [13]. In vivo (in humans), it is unknown whether the concentrations of 8-iso-PGF2α reached in cerebral arteries after aSAH are biologically relevant. The impact of elevated F2-IsoP levels on DCI remains to be established. Considering the relationship between 8-iso-PGF2α and clinical status, further work is needed to investigate the role of 8-iso-PGF2α, not just as a biomarker but also on its role in DCI onset.

To conclude, our results require further multicenter prospective studies to assess the optimal cut-off point for a DCI screening test. No doubt, research on oxidative stress and antioxidants warrants further investigation to apprehend the DCI pathophysiology.

## 5. Conclusions

We present a non-invasive biomarker that has a potential to be used in clinical practice as an additional diagnostic tool. In contrast to current diagnostic approaches, the proposed technique is not based on a complex model comprising numerous markers, but on a single measurement that may be directly interpreted on its own. This probably could, in practice, allow for a minimally invasive monitoring of DCI and provide an early warning should there be a need for prompt introduction of induced hypertension treatment and/or endovascular therapy, e.g., endovascular administration of verapamil [53]. To date, we have not investigated the correlation between induced hypertension treatment and F2-IsoP levels; however, we hope that this might be an interesting point of our future research.

## Figures and Tables

**Figure 1 diagnostics-11-00005-f001:**
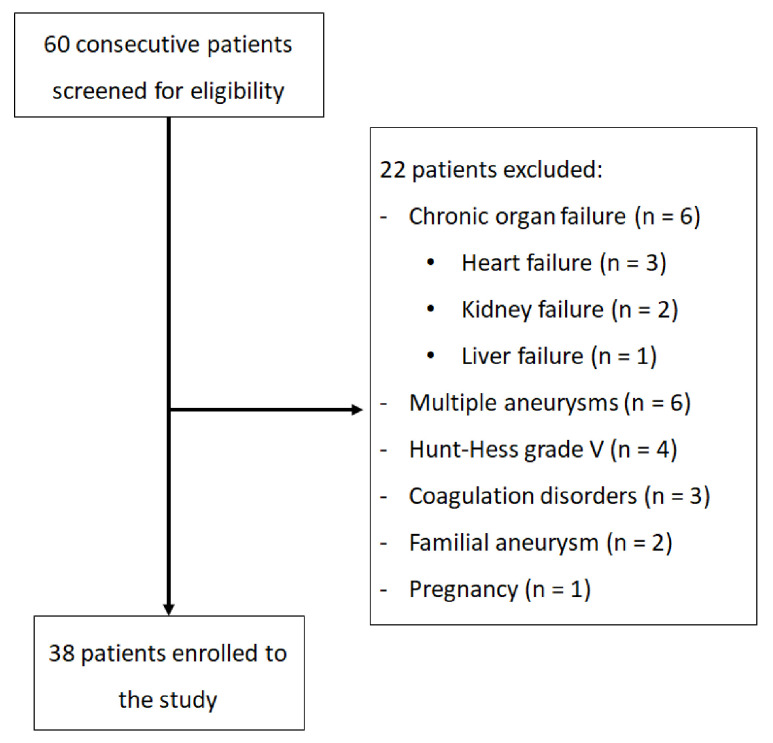
Flowchart presenting the enrollment of the patients.

**Figure 2 diagnostics-11-00005-f002:**
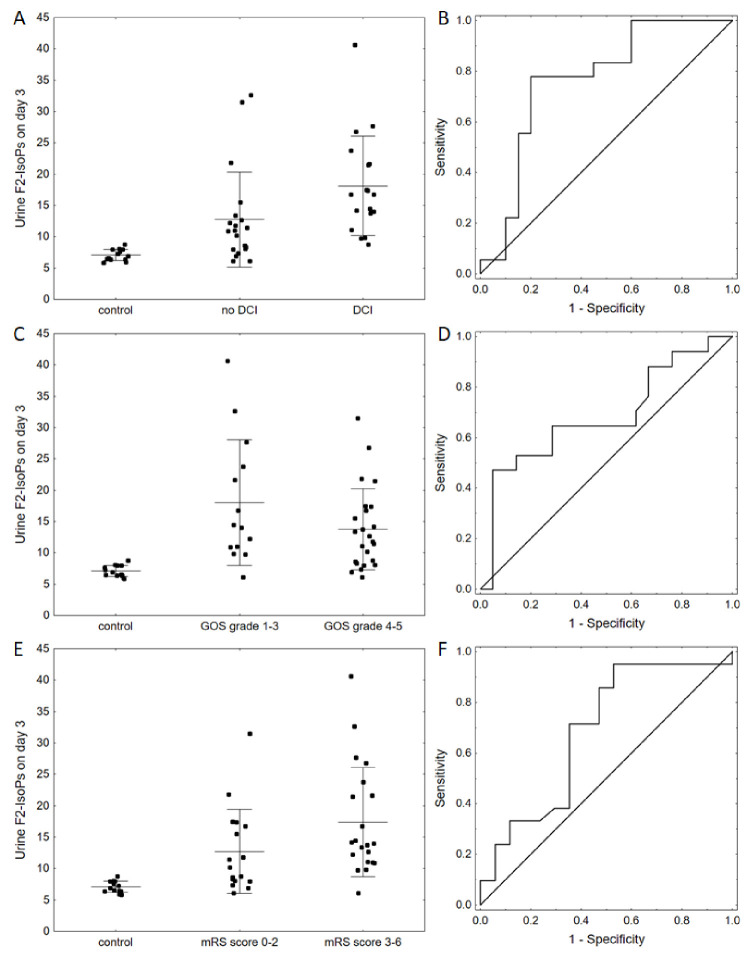
Differences in F2-isoprostane urine levels between different groups of patients; dots and lines show means ± SD (**A**) A plot for the control group and aneurysmal subarachnoid hemorrhage (aSAH) patients with and without delayed cerebral ischemia (DCI) (control vs. aneurysmal subarachnoid hemorrhage: *p* < 0.001 for ANOVA; *p* < 0.05 for all comparisons; DCI vs. no DCI: *p* = 0.023). (**B**) Receiver operating characteristic curve for DCI prediction by F2-isoprostane levels on day 3 (area under the curve (AUC) 0.764, 95% CI: 0.606–0.922; *p* = 0.001 optimal cutoff: threshold 13.8, sensitivity—0.778, specificity—0.800). (**C**) A plot for clinical condition of the patients 12 months after surgery (GOS scale) (*p* = 0.008). (**D**) Receiver operating characteristic curve for prediction of clinical condition after 1 and 12 months (GOS scale) by F2-isoprostane levels on day 3 (AUC 0.685, 95% CI: 0.507–0.863; *p* = 0.042 optimal cutoff: threshold 8.8, sensitivity—0.471, specificity—0.952). (**E**) A plot for clinical condition of the patients 12 months after surgery (mR scale) (*p* < 0.001 for ANOVA, *p* < 0.05 for all comparisons; DCI vs. no DCI *p* = 0.042). (**F**) Receiver operating characteristic curve for prediction of clinical condition after 12 months (mRS) by F2-isoprostane levels on day 3 (AUC 0.685, 95% CI 0.507–0.863; *p* = 0.042 optimal cutoff: threshold 9.8, sensitivity—0.952, specificity—0.471).

**Figure 3 diagnostics-11-00005-f003:**
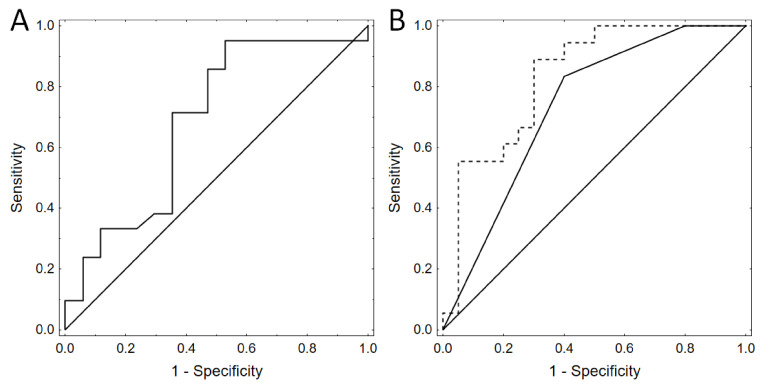
Receiver operating characteristic curves based on backward stepwise logistic regression showed that the best model for delayed cerebral ischemia (DCI) prediction based on clinical markers contained only Hunt and Hess scale (AUC 0.733, 95% CI: 0.572–0.894). When we added to this the model F2-IsoP levels on day 3, we received better results—AUC 0.833, 95% CI: 0.703–0.964. Thus, we improved the outcome prediction (*p* = 0.043). The differences remained significant after fivefold cross validation. (**A**) Receiver operating characteristic curves based on backward stepwise logistic regression models designed with resampling and cross-validation procedure. Prediction of delayed cerebral ischemia (DCI) by Hunt and Hess scale alone. (**B**) Receiver operating characteristic curves based on backward stepwise logistic regression models designed with resampling and cross-validation procedure. Dashed line—prediction of delayed cerebral ischemia (DCI) by Hunt and Hess scale and F2-IsoP levels on day 3; solid line—Hunt and Hess scale alone.

**Table 1 diagnostics-11-00005-t001:** Inclusion and exclusion criteria.

	Study Group	Control Group
Inclusion criteria	age between 18–75 years; single ruptured saccular intracranial aneurysm; treatment at the Department of Neurosurgery with microsurgery/endovascular technique; diagnosis and treatment within 1 day after aneurysm rupture; Hunt–Hess grade between I and IV.	age between 18–75; no aneurysm history.
Exclusion criteria	age <18 or >75 years; multiple or familial aneurysms; no surgical or endovascular treatment; diagnosis/treatment >1 day after aneurysm rupture; Hunt–Hess grade V; chronic organ failure: heart (NYHA III/IV), kidney (GFR < 30 mL/1.73 m^2^/min), liver (Child–Pugh score C/D); pregnancy; coagulation disorders; legal incapacitation.	age <18 or >75 years; chronic organ failure: heart (NYHA III/IV), kidney (GFR < 30 mL/1.73 m^2^/min), liver (Child–Pugh score C/D); pregnancy; coagulation disorders; legal incapacitation.

**Table 2 diagnostics-11-00005-t002:** Complete clinical data.

			Admission *			Discharge ^†^	1 Month ^‡^	12 Months ^§^			F2-IsoPs				
Case	Age (years)	Sex	H/H	Fisher	GOS	GOS	GOS	GOS	mRS	DCI (day)	Day 1	Day 2	Day 3	Day 4	Day 5
1	73	M	2	4	3	1	1	1	6	Yes	13.3	14.1	27.7	20.2	17
2	64	M	3	4	2	2	2	2	5	Yes	20.2	22.5	23.8	7.4	0.7
3	60	F	3	2	3	3	3	3	4	Yes	11.5	14.1	9.8	7.4	10.2
4	48	M	3	2	3	3	3	3	4	No	19.8	7.4	6.1	11.7	8
5	82	F	1	2	3	3	3	3	3	No	8.7	9.3	12.3	13.7	25.2
6	57	F	3	3	3	4	4	4	2	No	29.2	39.4	31.5	23.3	17.4
7	66	F	3	2	4	4	4	4	2	No	N/A	16.3	7.0	5.6	23.4
8	45	F	2	4	4	4	4	5	1	No	24.1	13.1	15.5	13.1	N/A
9	36	M	3	3	2	3	4	4	3	No	17	12.9	13.4	9.9	17.6
10	65	F	2	4	4	4	4	4	2	No	N/A	5.6	6.2	8.5	4.2
11	61	F	2	4	4	4	4	4	2	No	20.5	16.9	21.8	19.6	24.7
12	59	M	3	3	3	4	4	4	2	No	20.7	12	8.4	8.8	10.4
13	60	M	3	4	3	2	2	2	5	Yes	12.7	11.9	14	13.2	18.3
14	27	F	1	1	4	4	4	4	2	No	12.2	11	8.6	19.3	N/A
15	87	F	3	4	2	2	2	2	5	Yes	8.7	8	16.8	5.8	9.5
16	61	F	3	4	3	4	4	4	2	Yes	86.5	16	17.4	16.2	19.9
17	73	F	3	3	3	3	4	4	3	Yes	16.3	14.9	13.8	9.3	6
18	65	F	3	4	3	3	4	4	3	Yes	24.5	30.5	21.5	13.6	18.8
19	39	M	1	2	4	4	4	4	2	No	26.9	14.9	8.1	7.7	6.2
20	47	M	2	4	3	4	4	4	2	Yes	17.5	20	17.5	20.8	22
21	52	M	3	4	3	4	4	4	2	Yes	19.8	23.1	16.8	14	14.3
22	35	F	3	4	3	3	4	4	3	Yes	38.4	26.2	26.8	26.8	34.5
23	68	M	3	4	3	2	2	2	5	Yes	31.2	37	40.7	36.5	22.9
24	60	F	3	4	2	2	1	1	6	Yes	12.3	36.2	14.5	19	11.6
25	60	F	3	3	3	3	4	4	3	Yes	11.8	12	11.1	16.7	9.9
26	90	M	2	3	4	3	1	1	6	Yes	11.5	9.9	9.9	9.3	11.4
27	79	F	3	3	3	4	4	4	2	Yes	15.1	9.6	8.8	12.7	N/A
28	62	F	2	2	2	2	1	1	6	No	11.6	13	11	10.6	13.7
29	48	M	3	4	3	3	3	3	4	No	9.5	10.5	10.9	9.6	9.2
30	47	M	3	4	3	1	1	1	6	Yes	N/A	12.6	21.7	15.1	N/A
31	86	F	3	3	3	3	4	4	3	No	9.8	N/A	12.7	10.3	9.5
32	65	F	1	1	4	5	5	5	0	No	15.5	11	10.2	13	9.8
33	68	F	3	3	4	3	4	4	3	Yes	8.5	9	14.2	11.5	9.5
34	80	F	3	2	2	1	1	1	6	No	32.7	37.4	32.7	20.8	N/A
35	46	F	2	3	4	4	5	5	1	No	8.5	9.7	11.5	7.8	6.8
36	42	F	2	2	4	4	5	5	0	No	10.7	30.5	8	7.4	10.3
37	44	F	2	1	4	4	5	5	1	No	6.8	9.3	11.8	9.6	8.9
38	43	M	2	2	4	4	4	4	2	No	8.7	6	7.4	4.9	6

H/H, Hunt and Hess; GOS, Glasgow Outcome Scale; mRS, modified Rankin Scale; DCI, delayed cerebral ischemia; F2-IsoPs, F2-isoprostanes (in pg/mg creatinine); M, male; F, female; N/A, not available. * Clinical assessment on admission. † Clinical assessment on discharge. ‡ Clinical assessment after 1 month. § Clinical assessment after 12 months.

**Table 3 diagnostics-11-00005-t003:** Levels of urine F2-isoprostanes (pg/mg creatinine) in specific subgroups.

	Day 3	Mean	Peak
SAH (*N* = 38)	15.3 ± 6.2	15.5 ± 1.6	18.6 ± 8.6
Female sex (*N* = 24)	14.8 ± 5.4	15.8 ± 2	19.2 ± 11
Male sex (*N* = 14)	16.1 ± 7.3	15 ± 1.5	17.6 ± 5
≤60 years old (*N* = 16)	14 ± 5.7	14.8 ± 1.7	17.9 ± 7
≥60 years old (*N* = 22)	16.2 ± 6.5	15.9 ± 1.7	19.1 ± 10
SAH—CVS (*N* = 18)	18 ± 5.9	17.5 ± 1.9	21.1 ± 11.2
SAH—no CVS (*N* = 20)	12.7 ± 5.1	13.6 ± 1.6	16.2 ± 6.7
GOS 12 months = 4/5 (*N* = 24)	13.7 ± 4.9	15.3 ± 2.2	20.4 ± 9.7
GOS 12 months = 1/2/3 (*N* = 14)	17.9 ± 8	15.7 ± 1.6	17.9 ± 8
mRS 12 months = 0/1/2 (*N* = 17)	12.7 ± 5.1	21.5 ± 2.7	21.5 ± 10.7
mRS 12 months = 3/4/5/6 (*N* = 21)	17.4 ± 6.9	15.8 ± 1.4	17.5 ± 8.5
Anterior circulation bleeding (*N* = 33)	14.8 ± 6	15.2 ± 1.6	17.2 ± 8.1
Posterior circulation bleeding (*N* = 5)	18.2 ± 5.2	17.3 ± 4.4	27.5 ± 23.5
H/H = 1/2 (*N* = 15)	12.4 ± 4.3	12.9 ± 0.4	14 ± 4.9
H/H = 3 (*N* = 23)	17.1 ± 6.8	17.2 ± 3.2	21.7 ± 10.6
Fisher = 1/2 (*N* = 12)	11 ± 3.9	12.8 ± 1.7	15 ± 6.3
Fisher = 3/4 (*N* = 26)	17.2 ± 5.9	16.7 ± 1.8	20.3 ± 9.3
Controls (*N* = 13)	mean 7.1 ± 0.9		
Acute hydrocephalus group (*N* = 8)	mean 11.7 ± 0.5		
Intracerebral hemorrhage group (*N* = 5)	mean 11.8 ± 0.3		

Values are presented as mean (N); SAH, subarachnoid hemorrhage; CVS, cerebral vasospasm; GOS, Glasgow Outcome Scale; mRS, modified Rankin Scale; H/H, Hunt and Hess.

## Data Availability

Data available in a publicly accessible repository. The data presented in this study are openly available in open research data repository (Zenondo).

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
