# Peer review of "The Role of Urine F2-isoprostane Concentration in Delayed Cerebral Ischemia after Aneurysmal Subarachnoid Haemorrhage—A Poor Prognostic Factor"

_diagnostics, 2020, doi:10.3390/diagnostics11010005_

Round 1

Reviewer 1 Report

We read the work by Wisniewski et al with great interest. The authors present a prospective data collection of 38 SAH patients in which concentrations of the isoprostane variant, F2-isoprostane, was determined in morning urine. The hypothesis here being that during breakdown of subarachnoid blood, the release of free oxygen radicals triggers isoprostane formation and levels thereof might be reflective of the risk for delayed cerebral ischemia. The authors assessed urine concentrations during the first five days after onset of hemorrhage and used a cohort of 13 healthy individuals as a control group.

In response to their report, we have following remarks/comments:

Major comments

The terminology of cerebral vasospasm, symptomatic vasospasm, delayed cerebral ischemia (DCI) and delayed ischemic neurological deficits (DIND) are used and described as different entities. The consensus terminology covering the whole concept is in our opinion nowadays, delayed cerebral ischemia (DCI). This to bypass the confusion with angiographic vasospasm one the one hand and on the other hand emphasizing the multifactorial nature of the ongoing process were angiographic vasospasm is only a single culprit among many others. We therefore recommend the used of DCI as the correct terminology, especially as your applied clinical definition fits the broadly accepted definition by Vergouwen et al.

The sensitivity of isoprostanes for cerebral pathology should be discusses. Might an increase in more severely ill SAH cases be a reflection of general/systemic disease?

Are all 38 included patients consecutive cases. If so please provide a flow-chart to report the inclusion process transparently and so excluded cases can be identified.

Although not an exclusion criteria, is there a specific reason why no H&H grade 4 hemorrhages were included? Please address this.

Please provide specifics on how clinical outcome was assessed. Where there regular routine checkups, telephone interviews, where assessors blinded to additional patient data? Etc.

Please clarify why a urine analysis was used. All ICU treated SAH patients have some sort of vascular access so less invasiveness is not an argument. Is this type of isoprostane only detectable in urine? Please elucidate to the reader. Plasma/Serum levels would bypass the need for evening out concentrations based on creatinine measured urine density.

Why was a stepwise regression analysis preferred for a dichotomized clinical outcome variable instead of a more common (and less controversial) binomial logistic regression. Please clarify.

Line 184-185: The mean level of urine F2-IsoPs in the acute hydrocephalus group was elevated versus controls.

When speaking of mean levels compared between subgroups, what time frame is meant? First day, all post-hemorrhage days?

Please consistently provide a measure of central tendency accompanied by its measure of dispersion. The latter is often mission.

Minor comments

Please specify why cases with severe comorbidity such as organ failure were excluded. Does these diseases have a proven effect on isoprostane levels? This is not a problem but the reason why, should be reported.

Line 22: “we have studied F2-21 IsoPs (especially 8-iso PGF2a)”

                Please write abbreviations in full upon first mention

Line 35 – 37: “Worldwide, aneurysmal subarachnoid hemorrhage (aSAH) and its complication - cerebral vasospasm (CVS), kill or seriously debilitate roughly 1.2 million people annually”

                Please refer her to the appropriate epidemiological study

Line 41 – 42: “The pathophysiology of CVS, remains unclear [3]”

Please refer to an essential review paper concerning DCI pathogenesis.
e.g. Nat Rev Neurol. 2014 Jan;10(1):44-58. doi: 10.1038/nrneurol.2013.246. PMID: 24323051

Line 42 – 43: “Experimental studies have shown that the presence of blood in the subarachnoid 42 space triggers vasospasm (more blood, greater CVS risk). “

Please rephrase in a more refined manner. E.g. …the risk of DCI correlates with clinical SAH severity and radiographic blood load.

Line 44: “Currently, no reliable diagnostic tools are available in daily clinical practice.”

This phrase contradicts what comes thereafter. Should this mean: “No reliable diagnostic tool exists evaluating patients who are not clinically assessable.

Line 47 – 54: Please make not of the fact that DCI pathophysiology exceeds the causal link between angiographic vasospasm, DCI-related infarction and unfavorable outcome. Additional factors such as early brain injury, microvasospasm, microthrombus formation, cortical spreading depolarization, blood-brain barrier disruption and dysregulated autoregulation are at play. Either the multifactorial pathophysiology is addressed here, or even better, the effect of isoprostanes on these processes should be discussed in the discussion.

Line 60 – 61; “In the literature, it was demonstrated that 8-iso 60 PGF2a exhibit the strongest vasoconstricitive action. “

                Of what? Of all identified isoprostanes?

Line 68 – 69: “The project at both clinical and laboratory part was conducted according to the principles of the declaration of Helsinki”

                Please rephrase as the preposition “at” is used incorrectly.

Line 74 – 75: “Our study is scientifically justified. The study protocol was approved by the Bioethics Committee.”

                Please add/specify the institution/affiliation.

Line 79: “Informed consent was obtained from all individual participants included in the study.”

Also for higher-grade SAH cases? At time of inclusion? Was the next of kin contacted for unconscious patients.

Line 82: We performed a prospective analysis of 38 patients after aSAH.

This should result in 5 sample for 38 patients or 190 measurements. Please provide transparent information in the actual available number of usable samples.

Line 123 – 125: In our study CVS was diagnosed 123 in 19 patients between 4 and 8 days after SAH, by 2 independent neurosurgeons and a 124 radiologist.

                Please correct the typo, and transfer this sentence to the results section of the manuscript.

Line 191 – 192: “We observed that, patients clinical condition (according to GOS and mRS) was significantly correlated with Hunt and Hess grade and patient clinical condition after surgery. “

                This is ubiquitously proven and therefore not redundant in this analysis.

Line 203: “Please summarize table 1 and 3 (for example in a single sentence). The individual effects sizes do not offer any relevant additional information.

Line 212 – 213: Our statistical analysis showed that F2-IsoP levels on day 3 were associated with of CVS occurrence (P = 0.007) and poorer clinical condition after 1 month (P = 0.024) and 12 months 213 (P = 0.008). “

                Please remove “of”

Line 304 – 305 “Results of the existing studies on the pathophysiology of CVS are ambiguous and 304 difficult to explain. Despite this, several theories were proposed.”

                This sentence suggest you are referring to the current study. Better would be “Existing data on the pathophysiology…”.

Line 311 – 312: One of the redox switches relates to prostanoid synthesis (PGIS). As a key enzyme of PGIS, cyclooxygenase, is activated at low peroxide levels, oxidative stress 312 promotes isoprostane formation.

                We thought that isoprostane production is independent of COX? Is this not the case?

Line 320 – 321: We decided to quantify urinary F2-IsoPs in aSAH patients and to investigate their association with clinical conditions.

                Please specify clinical condition e.g. SAH severity and clinical outcome.

Line 373: “a possible need of prompt introduction of 3H treatment”

                Triple H treatment has come into disfavor due to the deleterious effects of hypervolemia and hemodilution. Is hyperdynamic treatment or induced hypertension meant?

Author Response

Reviewer #1 of Diagnostics,

I am pleased to submit the revised version of the manuscript entitled “The role of urine F2-isoprostane concentration in delayed cerebral ischemia after aneurysmal subarachnoid haemorrhage – a poor prognostic factor”. We highly appreciate your time and we would like to express our gratitude for the valuable criticism and suggestions. We addressed all your comments and performed a thorough language correction. All changes are highlighted in the revised manuscript. We also provide the point-by-point responses to your comments. We hope that our replies are satisfactory and that the revised version of the manuscript is suitable for publication in Diagnostics.

We look forward to your response.

Yours sincerely,

Karol Wiśniewski

Reviewer #1:

Major comments

  1. The terminology of cerebral vasospasm, symptomatic vasospasm, delayed cerebral ischemia (DCI) and delayed ischemic neurological deficits (DIND) are used and described as different entities. The consensus terminology covering the whole concept is in our opinion nowadays, delayed cerebral ischemia (DCI). This to bypass the confusion with angiographic vasospasm one the one hand and on the other hand emphasizing the multifactorial nature of the ongoing process were angiographic vasospasm is only a single culprit among many others. We therefore recommend the used of DCI as the correct terminology, especially as your applied clinical definition fits the broadly accepted definition by Vergouwen et al.

We thank for this remark; in the revised manuscript we employ the broad definition of DCI in line with Vergouwen et al..

  1. The sensitivity of isoprostanes for cerebral pathology should be discusses. Might an increase in more severely ill SAH cases be a reflection of general/systemic disease?

To address this issue we added the following paragraph to the discussion section:

“In the literature, elevated F2-IsoPs levels were reported in cerebral pathologies including chronic neurodegenerative diseases, including Alzheimer’s disease, Huntington’s disease, Parkinson’s disease and amyotrophic lateral sclerosis [36-40]. In addition, Millard et al. described an association between higher CSF levels of F2-IsoPs and poorer executive function during aging [41]. Their role was also studied in a small group of aSAH patients (n=15) with no observed relation to GOS, Fisher Scale or Hunt and Hess Grade scores [42]. Nonetheless, the sample size precludes any specific conclusions. “

  1. Are all 38 included patients consecutive cases. If so please provide a flow-chart to report the inclusion process transparently and so excluded cases can be identified.

60 consecutive aSAH patients were screened for eligibility. Among these, 38 fulfilled the eligibility criteria and were included in the study. The flowchart was added to the article.

  1. Although not an exclusion criteria, is there a specific reason why no H&H grade 4 hemorrhages were included? Please address this.

Patients with H&H grade 4 hemorrhages were not intentionally excluded from the study. Among the 60 patients screened for eligibility, there were 5 patients with H&H grade 4 hemorrhage. Each of the five patients met an exclusion criterion: two had multiple aneurysms, two had coagulation disorders and one had liver failure. We included this information in the article.

  1. Please provide specifics on how clinical outcome was assessed. Where there regular routine checkups, telephone interviews, where assessors blinded to additional patient data? Etc.

We assessed the outcome at discharge, 1 month and 12 months afterwards, according to Glasgow Outcome Scale (GOS) and the modified Rankin Scale (mRS). Outcome measures were obtained by research neurosurgeons blinded to the experimental results. Patients were scheduled for separate control visits apart from the out-patient clinic follow-up. In all possible cases we performed regular routine checkups which included full examination of the patient. The only exception were cases of patient's death, when we spoke with the family on the phone.

We included this information to the article.

  1. Please clarify why a urine analysis was used. All ICU treated SAH patients have some sort of vascular access so less invasiveness is not an argument. Is this type of isoprostane only detectable in urine? Please elucidate to the reader. Plasma/Serum levels would bypass the need for evening out concentrations based on creatinine measured urine density.

To address this issue we added the following paragraph to the discussion section:

“The choice of urine as the biomarker source was based on the properties of isoprostanes. They have a short blood half-life of about 16 minutes and are cleared primarily by the kidneys. In urine, the concentration 8-iso-PGF2α was shown to remain stable even after 5-day incubation of the sample at 37oC [43]. This makes urine the preferable non-invasive source for biomarker analysis.

  1. Why was a stepwise regression analysis preferred for a dichotomized clinical outcome variable instead of a more common (and less controversial) binomial logistic regression. Please clarify.

We are grateful for this remark and apologize for the mistake. Binominal logistic regression is exactly what had been done. The backward stepwise approach was used for feature selection. The mistake was corrected and the sentence rewritten, according to the Reviewer's remark.

  1. Line 184-185: The mean level of urine F2-IsoPs in the acute hydrocephalus group was elevated versus controls. When speaking of mean levels compared between subgroups, what time frame is meant? First day, all post-hemorrhage days? Please consistently provide a measure of central tendency accompanied by its measure of dispersion. The latter is often mission.

We thank the Reviewer for this remark. It is the mean level of all post-hemorrhage days. In the previous version of the manuscript, by mistake, we presented median values is Table 3. This is now corrected and Table 3 presents means ± standard deviation. We added the standard deviation values wherever it was missing.

Minor comments

  1. Please specify why cases with severe comorbidity such as organ failure were excluded. Does these diseases have a proven effect on isoprostane levels? This is not a problem but the reason why, should be reported.

Free F2-IsoPs are filtrated through kidneys and can be found in urine. Although this is the main mechanism of their elimination, F2-IsoPs can also be metabolised in the liver to a variety of compounds. An association between F2-IsoPs and chronic heart failure in humans was reported in the literature. Thus, we decided to eliminate systemic diseases which could potentially affect F2-IsoPs concentration. We added this information to the article.

  1. Line 22: “we have studied F2-21 IsoPs (especially 8-iso PGF2a)”. Please write abbreviations in full upon first mention.
  2. Line 35 – 37: “Worldwide, aneurysmal subarachnoid hemorrhage (aSAH) and its complication - cerebral vasospasm (CVS), kill or seriously debilitate roughly 1.2 million people annually”. Please refer her to the appropriate epidemiological study.
  3. Line 41 – 42: “The pathophysiology of CVS, remains unclear [3]”. Please refer to an essential review paper concerning DCI pathogenesis.
    g. Nat Rev Neurol. 2014 Jan;10(1):44-58. doi: 10.1038/nrneurol.2013.246. PMID: 24323051
  4. Line 42 – 43: “Experimental studies have shown that the presence of blood in the subarachnoid 42 space triggers vasospasm (more blood, greater CVS risk)“. Please rephrase in a more refined manner. E.g. …the risk of DCI correlates with clinical SAH severity and radiographic blood load.
  5. Line 44: “Currently, no reliable diagnostic tools are available in daily clinical practice.” This phrase contradicts what comes thereafter. Should this mean: “No reliable diagnostic tool exists evaluating patients who are not clinically assessable.

We apologize for the errors described in comments 10-14, the corrections are made as requested.

  1. Line 47 – 54: Please make not of the fact that DCI pathophysiology exceeds the causal link between angiographic vasospasm, DCI-related infarction and unfavorable outcome. Additional factors such as early brain injury, microvasospasm, microthrombus formation, cortical spreading depolarization, blood-brain barrier disruption and dysregulated autoregulation are at play. Either the multifactorial pathophysiology is addressed here, or even better, the effect of isoprostanes on these processes should be discussed in the discussion.

To address this issue we added the following paragraph to the discussion section:

“In line with the multifactorial DCI mechanisms, not only do isoprostanes promote vasoconstriction and microvasospasm, but also cause oxidative damage to the cell membrane or organelles. These compounds affect the fluidity and integrity of phospholipid membranes by changing the intermolecular interactions, leading to cell apoptosis and early brain injury. In addition, isoprostanes may stimulate platelet aggregation, promoting microthrombi formation, as well as induce cell cycle progression. Despite their clinical significance, no specific isoprostane receptor or the related intracellular pathway has been documented to date.”

  1. Line 60 – 61; “In the literature, it was demonstrated that 8-iso 60 PGF2a exhibit the strongest vasoconstricitive action. “ Of what? Of all identified isoprostanes?

The sentence was corrected.

  1. Line 68 – 69: “The project at both clinical and laboratory part was conducted according to the principles of the declaration of Helsinki”. Please rephrase as the preposition “at” is used incorrectly.
  2. Line 74 – 75: “Our study is scientifically justified. The study protocol was approved by the Bioethics Committee.” Please add/specify the institution/affiliation.
  3. Line 79: “Informed consent was obtained from all individual participants included in the study.”Also for higher-grade SAH cases? At time of inclusion? Was the next of kin contacted for unconscious patients.

In relation to points 17-19, we rephrased the whole section for better clarity:

“The study protocol was approved by the Bioethics Committee of the Medical University of Lodz (approval no. RNN/119/15/KE). The study was designed in accordance with the Good Clinical Practice (GCP) guidelines and was conducted according to the principles of the Declaration of Helsinki. An informed consent was obtained from the participants prior to inclusion. In case of depressed level of consciousness, the patient’s legal representative was asked for the informed consent.”

  1. Line 82: We performed a prospective analysis of 38 patients after aSAH. This should result in 5 sample for 38 patients or 190 measurements. Please provide transparent information in the actual available number of usable samples.

To address this issue we added the following paragraph to the methods section:

“There were 181 samples in the study group (out of the intended 190; nine samples were lost due to collection or storage error) and 13 samples in the control group available for measurements.”

  1. Line 123 – 125: In our study CVS was diagnosed 123 in 19 patients between 4 and 8 daysafter SAH, by 2 independent neurosurgeons and a 124 radiologist. Please correct the typo, and transfer this sentence to the results section of the manuscript.

We corrected the typo and transferred the sentence into the Results section.

  1. Line 191 – 192: “We observed that, patients clinical condition (according to GOS and mRS) was significantly correlated with Hunt and Hess grade and patient clinical condition after surgery. “This is ubiquitously proven and therefore not redundant in this analysis.

We removed this paragraph. We also removed table 2, which presented the correlation between Hunt and Hess grade and clinical condition of the patients after surgery.

  1. Line 203: “Please summarize table 1 and 3 (for example in a single sentence). The individual effects sizes do not offer any relevant additional information.

We removed the tables and summarized the urine F2-IsoPs levels in aSAH patients.

  1. Line 212 – 213: Our statistical analysis showed that F2-IsoP levels on day 3 were associated withof CVS occurrence (P = 0.007) and poorer clinical condition after 1 month (P = 0.024) and 12 months 213 (P = 0.008). “ Please remove “of”
  2. Line 304 – 305 “Results of the existing studies on the pathophysiology of CVS are ambiguous and 304 difficult to explain. Despite this, several theories were proposed.” This sentence suggest you are referring to the current study. Better would be “Existing data on the pathophysiology…”.

In relation to points 24-25, the requested corrections were made.

  1. Line 311 – 312: One of theredox switches relates to prostanoid synthesis (PGIS). As a key enzyme of PGIS, cyclooxygenase, is activated at low peroxide levels, oxidative stress 312 promotes isoprostane formation. We thought that isoprostane production is independent of COX? Is this not the case?

If the levels of peroxide are low, cyclooxygenase is activated (as under physiological conditions - without oxidative stress). In contrast, if the peroxide levels are high (as in oxidative stress), cyclooxygenase is not activated. Thus, oxidative stress promotes F2-IsoPs formation as a result of free-radicals peroxidation.

We rephrased the sentence:

“As a key enzyme of PGIS, cyclooxygenase, is activated at low peroxide levels. Oxidative stress (a condition with high peroxide levels) promotes isoprostane formation and inhibits PGIS synthesis”

  1. Line 320 – 321: We decided to quantify urinary F2-IsoPs in aSAH patients and to investigate their association with clinical conditions. Please specify clinical condition e.g. SAH severity and clinical outcome.

The requested correction was made.

  1. Line 373: “a possible need of prompt introduction of 3H treatment”. Triple H treatment has come into disfavor due to the deleterious effects of hypervolemia and hemodilution. Is hyperdynamic treatment or induced hypertension meant?

The sentence was corrected:

“This probably could, in practice, allow for a minimally invasive monitoring of DCI and provide an early warning should there be a need for prompt introduction of induced hypertension treatment and/or endovascular therapy, e.g. endovascular administration of verapamil [53].”

Reviewer 2 Report

In my opinion paper ,,The role of urine F2-isoprostane concentration in cerebral vasospasm after aneurysmal subarachnoid haemorrhage - a poor prognostic factor” presented to the review concerns on relevant topic in the management of aneurysmal subarachnoid hemorrhage complications. The elaborated material is original but minor, so the usefulness of the urine F2-isoprostane concentration measurements might be of limited clinical use. In my feeling perhaps too little attention has been paid to explain and distinguish the definition of radiological vasospasm and delayed cerebral ischemia, which is of more clinical character. Another issue is the way the researchers monitor patients for the radiological vasospasm – the schedule for this monitoring wasn’t clear. The title of the article is not coherent with its content. Reviewed paper is well structured. Cited literature contains the most important positions. I have acquainted myself with the results of this work with a great interest. I believe that this work is worthy of publication after revisions.

Author Response

Reviewer #2 of Diagnostics,

I am pleased to submit the revised version of the manuscript entitled “The role of urine F2-isoprostane concentration in delayed cerebral ischemia after aneurysmal subarachnoid haemorrhage – a poor prognostic factor”. We highly appreciate your time and we would like to express our gratitude for the valuable criticism and suggestions. We addressed all your comments and performed a thorough language correction. All changes are highlighted in the revised manuscript. We also provide the point-by-point responses to your comments. We hope that our replies are satisfactory and that the revised version of the manuscript is suitable for publication in Diagnostics.

We look forward to your response.

Yours sincerely,

Karol Wiśniewski

In my opinion paper ,,The role of urine F2-isoprostane concentration in cerebral vasospasm after aneurysmal subarachnoid haemorrhage - a poor prognostic factor” presented to the review concerns on relevant topic in the management of aneurysmal subarachnoid hemorrhage complications. The elaborated material is original but minor, so the usefulness of the urine F2-isoprostane concentration measurements might be of limited clinical use. In my feeling perhaps too little attention has been paid to explain and distinguish the definition of radiological vasospasm and delayed cerebral ischemia, which is of more clinical character. Another issue is the way the researchers monitor patients for the radiological vasospasm – the schedule for this monitoring wasn’t clear. The title of the article is not coherent with its content. Reviewed paper is well structured. Cited literature contains the most important positions. I have acquainted myself with the results of this work with a great interest. I believe that this work is worthy of publication after revisions.

  1. In my feeling perhaps too little attention has been paid to explain and distinguish the definition of radiological vasospasm and delayed cerebral ischemia, which is of more clinical character
  2. Another issue is the way the researchers monitor patients for the radiological vasospasm – the schedule for this monitoring wasn’t clear.

To address both issues, we restructured the first paragraph of the introduction section and we added the information on vasospasm monitoring to the methods section:

“The most dangerous complication of aSAH is cerebral vasospasm (CVS) leading to delayed cerebral ischemia (DCI). The concept of CVS is complex; it can be diagnosed both angiographically and clinically. The narrowing of the contrast column in major cerebral arteries (i.e. radiologic or angiographic vasospasm) is detected on angiograms in about 70% of patients. In theory, cerebral vasoconstriction should lead to generalized ischemia. Nonetheless, the resultant deterioration of the clinical status and neurological deficits (i.e. clinical vasospasm) can be observed in up to 25% of patients. Thus, the causal link between angiographic and clinical CVS may be often not observed in clinical practice. DCI is a broad term encompassing both symptomatic vasospasm and neurological deficits. Its pathophysiology is multifactorial and not entirely clear [2,3], exceeding the simple causal link between vasospasm and infarction. In each case of DCI, a number of aspects should be considered, including early brain injury, microcirculatory constriction, microthrombosis, cortical spreading depolarization, blood-brain barrier disruption, dysregulated autoregulation and delayed cell apoptosis.”

“DCI was surmised in patients with neurological deterioration (confusion or decreased level of consciousness by at least 1 point on the Glasgow Coma Scale, with or without focal neurologic deficits, lasting for at least 1 hour) after excluding other causes of neurologic deficits. Each time we suspected DCI, we performed transcranial Doppler (TCD) in search of radiological signs of vasospasm. We examined the flow velocity in the middle cerebral artery (MCA) and internal cerebral artery (ICA). If the flow velocity in MCA exceeded 120 cm/second and the Lindegaard ratio (MCA low velocity/ICA flow velocity) was greater than 3, we performed a digital subtraction angiography (DSA). In our opinion TCD is not sufficient as a single screening measure. DSA is the gold standard and offers the possibility of endovascular treatment.”

  1. The title of the article is not coherent with its content.

To address this issue, we changed the title to: “The role of urine F2-isoprostane concentration in delayed cerebral ischemia after aneurysmal subarachnoid haemorrhage – a poor prognostic factor”

Round 2

Reviewer 1 Report

We believe all appropriate changes have been made. 

Kind regards,

Dr. med. Michael Veldeman

Reviewer 2 Report

After making corrections, the title of the article better reflects its content. The introduction to the work looks much better and is understandable. The differentiation between CVS and DCI and the relationship between them are explained. In the "Material and Methods" section, the methodology and diagnosis of DCI among the studied patients is better described. The slightly extended discussion should certainly bring out even more clearly the value of this article.
The changes contributed to a significant improvement in the quality of the publication.